

# Latitudinal variation in phlorotannin contents from Southwestern Atlantic brown seaweeds

Glaucia Ank[1], Bernardo Antônio Perez da Gama[1] and Renato Crespo Pereira[1,2]

[1] Departamento de Biologia Marinha, Universidade Federal Fluminense, Niterói, Rio de Janeiro, Brazil
[2] Instituto de Pesquisas Jardim Botânico do Rio de Janeiro, Rio de Janeiro, Brazil

## ABSTRACT

Phlorotannins are primary and/or secondary metabolites found exclusively in brown seaweeds, but their geographic distribution and abundance dynamic are not very well understood. In this study we evaluated the phlorotannin concentrations among and within-species of brown seaweeds in a broad latitudinal context (range of 21°) along the Brazilian coast (Southwestern Atlantic), using the Folin-Ciocalteau colorimetric method. In almost all species (16 out of 17) very low phlorotannin concentrations were found (<2.0%, dry weight for the species), confirming reports of the typical amounts of these chemicals in tropical brown seaweeds, but with significantly distinct values among seven different and probably highly structured populations. In all 17 seaweed species (but a total of 25 populations) analyzed there were significant differences on the amount of phlorotannins in different individuals ($t$-test, $p < 0.01$), with coefficients of variation (CV) ranging from 5.2% to 65.3%. The CV, but not the total amount of phlorotannins, was significantly correlated with latitude, and higher values of both these variables were found in brown seaweeds collected at higher latitudes. These results suggest that brown seaweeds from higher latitudes can produce phlorotannins in a wider range of amounts and probably as response to environmental variables or stimuli, compared to low latitude algae.

Subjects Ecology, Marine Biology
Keywords Phlorotannins, Latitudinal trend, Phaeophyceae, Tropical seaweeds

## INTRODUCTION

Phlorotannins are polymers derived from a simple monomer, phloroglucinol, found exclusively in brown seaweeds (*Targett & Arnold, 1998*, *2001*). These water-soluble secondary metabolites constitute a special class of polyphenols that may exhibit multifunctional ecological roles, acting as a herbivore deterrent (*Pereira & Yoneshigue-Valentin, 1999*), antifouling agent (*Plouguerné et al., 2012*), antioxidant (*Cruces, Huovinen & Gómez, 2012*), UV protector (*Henry & Van Alstyne, 2004*), and a chelating agent of toxic heavy metal ions (*Karez & Pereira, 1995*). However, these chemicals may also be classified as primary metabolites when they are structural components of cell walls (*Schoenwaelder & Clayton, 1999*). In fact, phlorotannins found inside the cells of brown seaweeds are stored in small vesicles called physodes, and these chemicals may exude into

Corresponding author
Renato Crespo Pereira,
rcrespo@id.uff.br

the environment due to their water solubility (*Jennings & Steinberg, 1994*) where they can have several vital ecological roles (*Pereira et al., 1990*). As cell wall components, where they form a complex with alginic acid, they are insoluble (*Schoenwaelder, 2002*; *Koivikko et al., 2005*). Given the smaller amounts of cell-wall-bound phlorotannins compared to soluble phlorotannins, the major function of these chemicals appears to be secondary metabolites (*Koivikko et al., 2005*).

The concentration of phlorotannins in brown seaweeds is known to be highly variable in several modes and at various scales, supposedly in response to the dynamics of biotic and abiotic environmental conditions (*Jormalainen et al., 2003*). For example, concentrations may vary in response to environmental factors, either biotic—such as herbivory (*Hemmi, Honkanen & Jormalainen, 2004*) and epibiosis (*Plouguerné et al., 2010*)—or abiotic—such as temperature (*Cruces, Huovinen & Gómez, 2012*), irradiance (*Cruces, Huovinen & Gómez, 2013*), nitrogen concentrations (*Pavia & Toth, 2000*), bathymetric variation, and immersion time in the intertidal range (*Connan et al., 2004*). Phlorotannin content can also vary according to intrinsic aspects of brown seaweeds, such as individual size and age (*Pavia et al., 2003*), and tissue type (*Plouguerné et al., 2012*).

Another interesting aspect relating to the distribution, abundance, and function of phlorotannins is the latitudinal differences in content of these chemicals among brown seaweeds living along large temperate-tropical gradients (*Steinberg, 1989*; *Van Alstyne & Paul, 1990*). High concentrations of these compounds have been found in species from high latitudes (*Ragan & Glombitza, 1986*; *Steinberg & Paul, 1990*; *Steinberg & Van Altena, 1992*; *Hay & Steinberg, 1992*; *Steinberg, 1992*). For example, species of Fucales and Laminariales that are abundant in temperate benthic communities, and Dictyotales found both in temperate and tropical regions, exhibit this biogeographic trend. The most common brown seaweed species in temperate Australasia exhibit more than 10% of total phlorotannins (*Steinberg, 1989*), whereas there are both phlorotannin-rich and -poor species in some temperate regions of South Africa (*Anderson & Velimirov, 1982*; *Tugwell & Branch, 1989*), northwestern Pacific (*Katayama, 1951*; *Estes & Steinberg, 1988*), and the European North Atlantic (*Ragan & Glombitza, 1986*).

Many species of brown seaweeds from North America exhibit low levels of phlorotannins, ranging from 0% to 2% of algal dry weight (DW) (*Ragan & Glombitza, 1986*). This range is found mainly in kelps dominating both the sublittoral and lower littoral environments (*Steinberg, 1992*). In contrast, as the most abundant organisms found in littoral and upper sublittoral regions, fucoids commonly contain higher phlorotannin contents (more than 4% DW) (*Steinberg, 1985*; *Van Alstyne, 1988*; *Denton, Chapman & Markham, 1990*; *Targett et al., 1992*). In general, brown seaweeds from North America exhibit broad variation in phlorotannin contents linked to the bathymetric gradient, with littoral fucoids and subtidal kelps showing high and low levels of these compounds, respectively (*Estes & Steinberg, 1988*; *Steinberg, 1992*).

In general, the intensity of selective pressures on organisms increases with decreasing latitude, including higher herbivory and epibiosis (*Railkin, 2004*; *Targett & Arnold, 1998*). Consequently, tropical seaweeds are hypothesized to have evolved more effective chemical defenses (*Van Alstyne & Paul, 1990*; *Targett et al., 1992*). Contrary to this trend,

phlorotannins are sometimes absent or present in very low concentrations in seaweeds from tropical environments (*Steinberg, 1989*; *Van Alstyne & Paul, 1990*; *Pereira & Yoneshigue-Valentin, 1999*). There is only one report of high amounts of these compounds in brown seaweeds from low latitudes (*Targett et al., 1995*).

However, in almost all studies, the quantification of phlorotannins is based on an analysis of distinct specimens of brown seaweed species extracted together, masking possible variability in amounts of these chemicals in each individual of a population. However, intra-populational variation in seaweed-derived chemicals can be of great magnitude and ecological significance (*Oliveira et al., 2013*).

Along the Brazilian coast, the few studies on phlorotannin contents in brown seaweeds are united in the fact that they typically reveal low concentrations (*Fleury et al., 1994*), and that they may be capable of inhibiting grazing when they occur at higher concentrations (*Pereira & Yoneshigue-Valentin, 1999*). The extensive Brazilian coast covers a broad latitudinal range of the Southwestern Atlantic and harbors numerous species of brown seaweeds. It comprises several environments suitable for exploring chemical defenses via a biogeographic approach. To date, most studies in Brazil have only reported average phlorotannin concentrations, so there is no information concerning the variation within populations or among populations from different latitudes. Thus, more in-depth analysis is needed, as tropical species could have the same mean value as temperate seaweeds, but exhibit greater standard deviation. Here, we hypothesized that contents of brown seaweed phlorotannins would exhibit latitudinal variation along the Brazilian coast. Our aim was to compare the mean phlorotannin concentration, as well as the coefficient of variation, among and within species of brown seaweeds across a broad latitudinal context along the Brazilian coast to evaluate the hypothesis that species from low latitudes exhibit lower amounts of these chemicals relative to those from high latitudes.

## MATERIALS AND METHODS

### Study organisms and collection localities

Brown seaweeds were collected from along the Brazilian coast (Instituto Chico Mendes de Conservação da Biodiversidade—Authorization Number 27001-2) in order to best represent various populations of the same species and individuals in each population from the different localities (Fig. 1; Table 1): Giz Beach (6°10′S; 35°05′W) at Tibau do Sul, RN; Itapuama (08°17′S; 34°57′W), Calhetas (08°20′S; 34°56′W), Paraíso (08°21′S; 34°57′W) and Suape beaches (08°22′S; 34°56′W) at Recife, PE; Itapuã Beach (12°57′S; 38°22′W) at Salvador, BA; Pé de Serra Beach (14°28′S; 39°01′W) at Uruçuca, BA; Morro de Pernambuco (14°48′S; 39°01′W) and Back Door beaches (14°56′S; 39°00′W) at Ilhéus, BA; Ponta Beach (16°24′S; 39°02′W) at Porto Seguro, BA; Três Praias (20°38′S; 40°28′W) at Guarapari, ES; Rasa (22°44′S; 41°57′W) and Forno beaches (22°45′S; 41°52′W) at Armação dos Búzios, RJ; and Canasvieiras Beach (27°25′S; 48°28′W) at Florianópolis, SC. We collected individuals of the following species: *Canistrocarpus cervicornis* (Kützing) De Paula & De Clerck, *Colpomenia sinuosa* (Mertens ex Roth) Derbés & Solier, *Dictyopteris delicatula* J.V. Lamouroux, *Dictyopteris polypodioides* (A.P. De Candolle) J.V. Lamouroux, *Dictyota ciliolata* Sonder ex Kützing, *Dictyota crispata* J.V. Lamouroux,

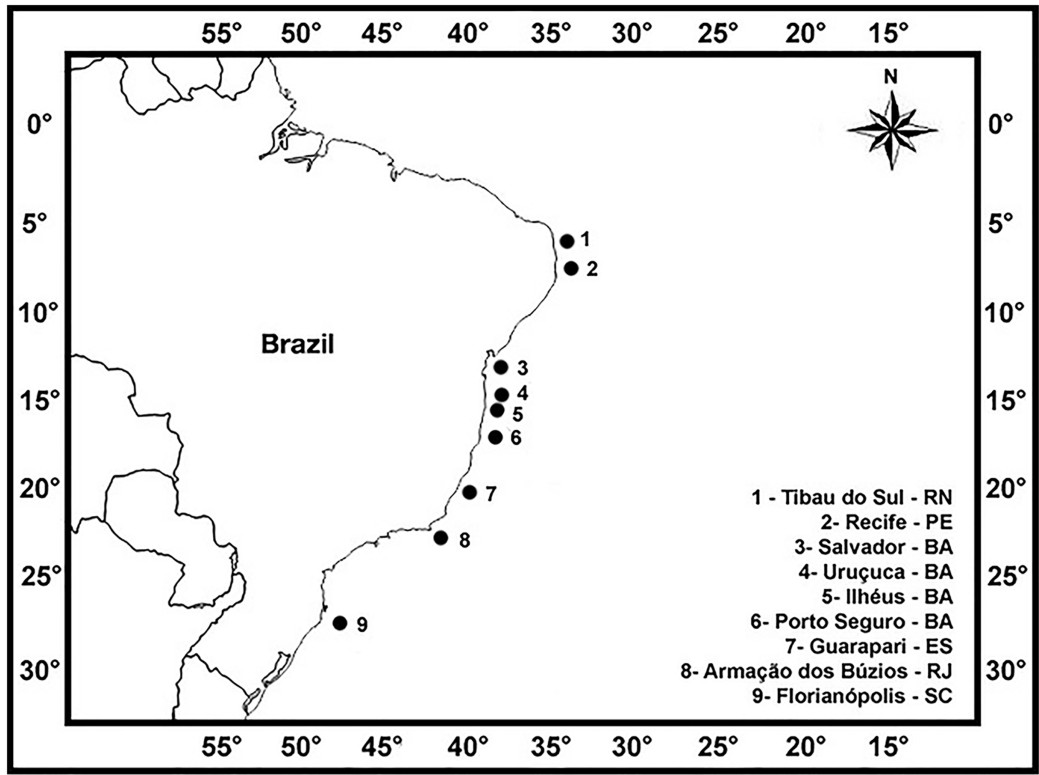

**Figure 1 Sampling sites.** Sampling sites of collect of the brown seaweeds studied along the Brazilian littoral, in a latitudinal range of 21°.

*Dictyota dichotoma* (Hudson) J.V. Lamouroux, *Dictyota mertensii* (Martius) Kützing, *Dictyota pfaffii* Schnetter, *Lobophora variegata* (J.V. Lamouroux) Womersley ex E.C. Oliveira, *Padina gymnospora* (Kützing) Sonder, *Sargassum filipendula* C. Agardh, *Sargassum stenophyllum* Martius, *Sargassum ramifolium* Kützing, *Sargassum vulgare* C. Agardh, *Sargassum vulgare* var. *nanum* E. De Paula, *Sargassum vulgare* var. *vulgare*, *Spatoglossum schroederi* (C. Agardh) Kützing, and *Stypopodium zonale* (J.V. Lamouroux) Papenfuss.

## Extraction

After collection, the seaweeds were freeze-dried, ground to powder and, before extraction, subjected to a lipid-removal treatment using one mL hexane for 3 min (*Koivikko et al., 2007*). Extraction was then carried out for 2 h using 10 mL of acetone:water (7:3) for 100 mg of each sample of dry alga. Each extract was centrifuged for 10 min at 3,500 rpm and filtered. Acetone was evaporated off at room temperature and the aqueous extract was again centrifuged. The supernatant was frozen for further quantification.

## Phlorotannin quantification

We used the Folin-Ciocalteau (FC) colorimetric method to quantify phlorotannin concentration, by which 1 N FC reagent was added to a diluted aliquot of the extract and, after 3 min, 20% sodium carbonate was added. After 45 min in the dark, phlorotannins were quantified in a Shimadzu UV1800 spectrophotometer, at 750 nm, using a

**Table 1 Brown seaweeds studied, number of specimens and corresponding collection places (x = The individuals were analyzed together because of the small size/biomass of the specimens; while in the remaining species, the analyzes were performed in each individual).**

| Seaweed species | Sampling sites | | | | | | | | |
|---|---|---|---|---|---|---|---|---|---|
| | 1. Tibau do Sul—RN | 2. Recife—PE | 3. Salvador—BA | 4. Uruçuca—BA | 5. Ilhéus—BA | 6. Porto Seguro—BA | 7. Guarapari—ES | 8. Búzios—RJ | 9. Florianópolis—SC |
| *Canistrocarpus cervicornis* | | 7 | | | | | | x | |
| *Colpomenia sinuosa* | | | | | | 4 | | x | |
| *Dictyopteris delicatula* | | 8 | 10 | | | x | x | | |
| *Dictyopteris polypodioides* | | | | | | | x | | |
| *Dictyota ciliolata* | | | | | | 11 | | | |
| *Dictyota crispata* | | | | | | 19 | | | |
| *Dictyota dichotoma* | | | | | | x | | | |
| *Dictyota mertensii* | | 7 | | | | | x | 10 | |
| *Dictyota pfaffii* | | | 10 | | | | | | |
| *Lobophora variegata* | | 10 | x | | | 10 | | | |
| *Padina gymnospora* | x | 10 | | x | x | 7 | x | x | 8 |
| *Sargassum filipendula* | | | 9 | 10 | | | | | |
| *Sargassum ramifolium* | | | | | | | 7 | | |
| *Sargassum stenophyllum* | | | | | | | | | 7 |
| *Sargassum vulgare* | | 15 | | 6 | 10 | 22 | | 5 | |
| *Spatoglossum schroederi* | | | | | | | 10 | | |
| *Stypopodium zonale* | | | | | | | | 32 | |

standard curve obtained with phloroglucinol ($r^2$ = 0.99), which is a monomer that absorbs under the same patterns as the polymers (phlorotannins) derived from it (*Steinberg, 1988*). Three aliquots of each extract were prepared for quantification, and the total phlorotannin concentration is expressed in % per DW of the seaweed.

## Statistical analysis

The coefficient of variation was calculated as the ratio of the standard deviation to the mean (coefficients of variation (CV) = δ/μ·100) in order to compare the amount of variation in phlorotannin contents observed within different populations of seaweeds. Total phlorotannin content of different populations from the same species was assessed

**Table 2 Number of individuals (N), and mean total phlorotannin content (TPC) measured in % (average ± standard deviation) of dry weight (DW) for the populations of seaweeds studied from different collection sites, including the coordinates, coefficient of variation (CV) and the ANOVA results for intra-populational variation (IV).**

| Seaweeds | Time of year | Location | Latitude (°S) | N | TPC (%DW) | IV | CV (%) |
|---|---|---|---|---|---|---|---|
| *C. cervicornis* | Spring/09 | 2 | 8 | 7 | 0.13 ± 0.01 | $F = 45.3; p < 0.001$ | 8.5 |
| *C. cervicornis* | Summer/11 | 9 | 22 | + | 0.18 ± 0.00 | + | + |
| *C. sinuosa* | Summer/11 | 7 | 16 | 4 | 0.07 ± 0.01 | + | + |
| *C. sinuosa* | Summer/11 | 9 | 22 | + | 0.24 ± 0.02 | + | + |
| *D. ciliolata* | Summer/11 | 7 | 16 | 11 | 0.14 ± 0.02 | $F = 126.3; p < 0.001$ | 13.9 |
| *D. crispata* | Summer/11 | 7 | 16 | 19 | 0.14 ± 0.04 | $F = 646.9; p < 0.001$ | 25.7 |
| *D. delicatula* | Spring/09 | 2 | 8 | 8 | 0.14 ± 0.01 | $F = 31.8; p < 0.001$ | 7.7 |
| *D. delicatula* | Summer/11 | 3 | 12 | 10 | 0.08 ± 0.01 | $F = 6.8; p < 0.001$ | 19.1 |
| *D. delicatula* | Summer/11 | 7 | 16 | + | 0.13 ± 0.01 | + | + |
| *D. delicatula* | Summer/11 | 8 | 20 | + | 0.12 ± 0.02 | + | + |
| *D. dichotoma* | Summer/11 | 7 | 6 | + | 0.11 ± 0.01 | + | + |
| *D. mertensii* | Spring/09 | 2 | 8 | 7 | 0.19 ± 0.01 | $F = 15.7; p < 0.001$ | 5.2 |
| *D. mertensii* | Summer/11 | 8 | 20 | + | 0.10 ± 0.01 | + | + |
| *D. mertensii* | Summer/11 | 9 | 22 | 10 | 0.18 ± 0.03 | $F = 73.9; p < 0.001$ | 15.4 |
| *D. pfaffii* | Summer/11 | 3 | 12 | 10 | 0.10 ± 0.02 | $F = 34.1; p < 0.001$ | 16.6 |
| *D. polypodioides* | Summer/11 | 8 | 20 | + | 0.22 ± 0.01 | + | + |
| *L. variegata* | Spring/09 | 2 | 8 | 10 | 0.91 ± 0.22 | $F = 366.0; p < 0.001$ | 24.1 |
| *L. variegata* | Summer/11 | 3 | 2 | + | 0.13 ± 0.00 | + | + |
| *L. variegata* | Summer/11 | 7 | 16 | 10 | 0.81 ± 0.53 | $F = 3765.0; p < 0.001$ | 65.3 |
| *P. gymnospora* | Autumn/11 | 1 | 6 | + | 0.40 ± 0.02 | + | + |
| *P. gymnospora* | Spring/09 | 2 | 8 | 10 | 0.07 ± 0.01 | $F = 58.1; p < 0.001$ | 13.1 |
| *P. gymnospora* | Summer/11 | 4 | 14 | + | 0.19 ± 0.01 | + | + |
| *P. gymnospora* | Summer/11 | 5 | 14 | + | 0.26 ± 0.02 | + | + |
| *P. gymnospora* | Summer/11 | 6 | 14 | + | 0.05 ± 0.00 | + | + |
| *P. gymnospora* | Summer/11 | 7 | 16 | 7 | 0.13 ± 0.05 | $F = 127.3; p < 0.001$ | 42.7 |
| *P. gymnospora* | Summer/11 | 8 | 20 | + | 0.09 ± 0.02 | + | + |
| *P. gymnospora* | Summer/11 | 9 | 22 | + | 0.22 ± 0.09 | + | + |
| *P. gymnospora* | Autumn/10 | 10 | 7 | 8 | 0.58 ± 0.30 | $F = 802.3; p < 0.001$ | 51.8 |
| *S. filipendula* | Summer/11 | 3 | 12 | 9 | 0.09 ± 0.00 | $F = 48.9; p < 0.001$ | 7.6 |
| *S. filipendula* | Summer/11 | 4 | 14 | 10 | 0.38 ± 0.10 | $F = 1166.8; p < 0.001$ | 25.6 |
| *S. ramifolium* | Summer/11 | 8 | 20 | 7 | 0.17 ± 0.06 | $F = 166.5; p < 0.001$ | 36.5 |
| *S. schroederi* | Summer/11 | 8 | 20 | 10 | 4.30 ± 0.78 | $F = 180.1; p < 0.001$ | 18.1 |
| *S. stenophyllum* | Autumn/10 | 10 | 27 | 7 | 0.45 ± 0.19 | $F = 109.9; p < 0.001$ | 42.0 |
| *S. vulgare* | Spring/09 | 2 | 8 | 15 | 0.13 ± 0.01 | $F = 85.8; p < 0.001$ | 9.6 |
| *S. vulgare* | Summer/11 | 4 | 14 | 6 | 0.14 ± 0.04 | $F = 4335.0; p < 0.001$ | 26.3 |
| *S. vulgare* | Summer/11 | 5 | 14 | 12 | 0.73 ± 0.15 | $F = 90.5; p < 0.001$ | 20.7 |
| *S. vulgare* | Summer/11 | 6 | 14 | 10 | 0.20 ± 0.11 | $F = 1428.1; p < 0.001$ | 53.9 |
| *S. vulgare* | Summer/11 | 7 | 16 | 10 | 0.10 ± 0.02 | $F = 89.9; p < 0.001$ | 18.1 |
| *S. vulgare* | Summer/11 | 9 | 22 | 5 | 1.10 ± 0.31 | $F = 212.8; p < 0.001$ | 30.9 |
| *S. zonale* | Summer/12 | 9 | 22 | 32 | 1.72 ± 0.49 | $F = 37.2; p < 0.001$ | 28.3 |

**Notes:**
1, Tibau do Sul; 2, Recife; 3, Salvador; 4, Uruçuca; 5, Ilhéus (Morro de Pernambuco); 6, Ilhéus (Back Door); 7, Porto Seguro; 8, Guarapari; 9, Armação dos Búzios; 10, Florianópolis.
+ Insufficient biomass for individual analysis.

by independent *t*-test or, when *n* was unequal, with an independent *t*-test with separate variances, which is more appropriate when considering groups of different sample sizes. In the case of more than two populations from the same species, we conducted a unifactorial ANOVA followed by the post-hoc Student Newman-Keuls test.

## RESULTS

### Amounts of phlorotannins and their inter-populational variability

Total phlorotannins ranged from 0.05% to 4.30% (average ± standard deviation) for the 17 brown seaweed species we studied (DW), encompassing a total of 25 populations (Table 2).

*Lobophora variegata* was the only species that did not show significant inter-populational variation, with mean phlorotannin contents of 0.91% (±0.22), 0.13% (±0.00) and 0.81% (±0.53) for the populations from Recife, Salvador and Porto Seguro, respectively ($p = 0.06$; $F(2.17) = 3.29$; ANOVA).

We found 0.13% (±0.01) and 0.18% (±0.00) of phlorotannins per DW of seaweed in the populations of *Canistrocarpus cervicornis* from Recife and Armação dos Búzios, respectively, with these values being significantly different ($p < 0.0001$; $t(8) = 6.75$; *t*-test for independent samples with separated variances). Populations of *Colpomenia sinuosa* collected at Porto Seguro and Armação dos Búzios had average phlorotannin contents of 0.07% (±0.01) and 0.24% (±0.02), respectively, with significant inter-populational variation ($p < 0.0001$; $t(5) = 16.62$; *t*-test for independent samples with separate variances).

In *Dictyopteris delicatula*, we recorded significant differences in the amounts of phlorotannins between the studied populations ($p = 0$; $F(3.19) = 45.76$; ANOVA). Individuals from Recife contained a mean phlorotannin content of 0.14% (±0.01), whereas specimens from Salvador, Porto Seguro and Guarapari had mean contents of 0.08% (±0.01), 0.13% (±0.01) and 0.12% (±0.02), respectively.

Individuals of *Dictyota mertensii* from three collection sites also contained significantly different phlorotannins contents ($p < 0.0001$; $F(2.24) = 16.71$; ANOVA). The Recife population exhibited a mean phlorotannin content of 0.19% (±0.01), whereas populations from Guarapari and Armação dos Búzios presented mean values of 0.10% (±0.01) and 0.18% (±0.03), respectively.

Populations of *Padina gymnospora* also differed in their mean phlorotannin contents ($p < 0.0001$; $F(8,34) = 9.78$; ANOVA), with the highest amount found in specimens from Florianópolis at 0.58% (±0.30). In the population of that same species from Tibau do Sul, we recorded 0.40% (±0.02) of phlorotannins per DW, whereas the mean value for the population from Recife was 0.07% (±0.01). At Ilhéus, the population from Uruçuca exhibited 0.19% (±0.01) phlorotannin content, whereas those from Morro de Pernambuco Beach and Back Door Beach had values of 0.26% (±0.02) and 0.05% (±0.00), respectively. Mean phlorotannin content of the population at Porto Seguro was 0.13% (±0.05), whereas it was 0.22% (±0.09) and 0.58% (±0.30) for those at Armação dos Búzios and Florianópolis, respectively.

We also observed significant variation in amounts of phlorotannins for populations of *Sargassum filipendula* ($p = 0$; $t(17) = −8.75$; *t*-test for independent samples with

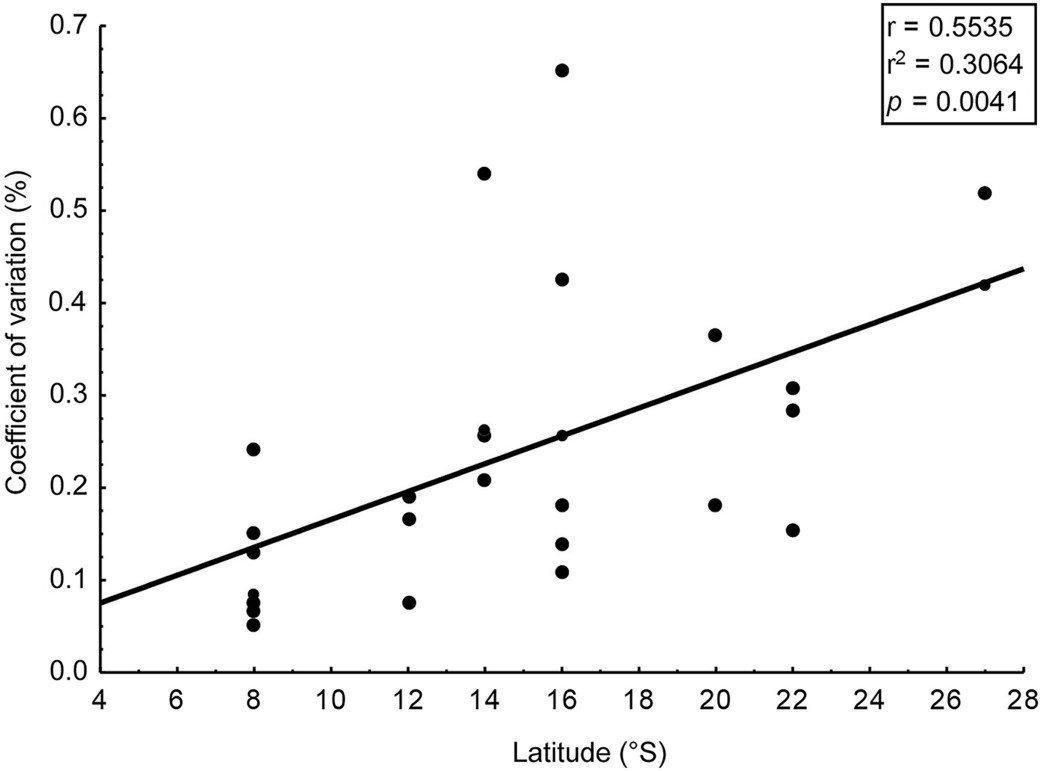

**Figure 2 Correlation: coefficient of variation (%) in content of phlorotannins and latitude.** Correlation between the coefficient of variation (%) in content of phlorotannins found in brown seaweeds and latitude (sampling site of the seaweeds).

separate variances), with individuals of the population from Salvador having significantly less phlorotannins 0.09% (±0.00) than those from Uruçuca 0.38% (±0.10). Similarly, populations of *Sargassum vulgare* were significantly different in terms of their phlorotannin contents ($p = 0$; $F(5.52) = 80.01$; ANOVA), with mean contents of 0.13% (±0.01) and 0.14% (±0.04) for the Recife and Uruçuca populations, respectively. Individuals of *Sargassum vulgare* from Morro de Pernambuco Beach (Ilheus) presented a phlorotannin content of 0.73% (±0.15), whereas specimens from Back Door Beach, also at Ilhéus, had 0.20% (±0.11). Exemplifying the diversity in phlorotannin contents, the population of *Sargassum vulgare* from Porto Seguro had the lowest value at 0.10% (±0.02) and the highest value was found for the population from Armação dos Búzios at 1.10% (±0.31).

Only one population was sampled for each of the following species: *Dictyopteris polypodioides* and *Sargassum ramifolium* from Guarapari showed a mean phlorotannin content of 0.22% (±0.01) and 0.17 (±0.06), respectively. For *Dictyota ciliolata* it was 0.14% (±0.02), for *Dictyota crispata* it was 0.14% (±0.04), for *Dictyota dichotoma* it was 0.11% (±0.01), all hailing from Porto Seguro. *Dictyota pfaffii* from Salvador—BA presented 0.10% (±0.02) of phlorotannins per DW, and for *Sargassum stenophyllum* from Florianópolis—SC it was 0.45% (±0.19). Mean phlorotannin content of *Stypopodium zonale* was 1.72% (±0.49), and the highest concentration of phlorotannins found for all studied species was in *Spatoglossum schroederi* at 4.30% (±0.78).

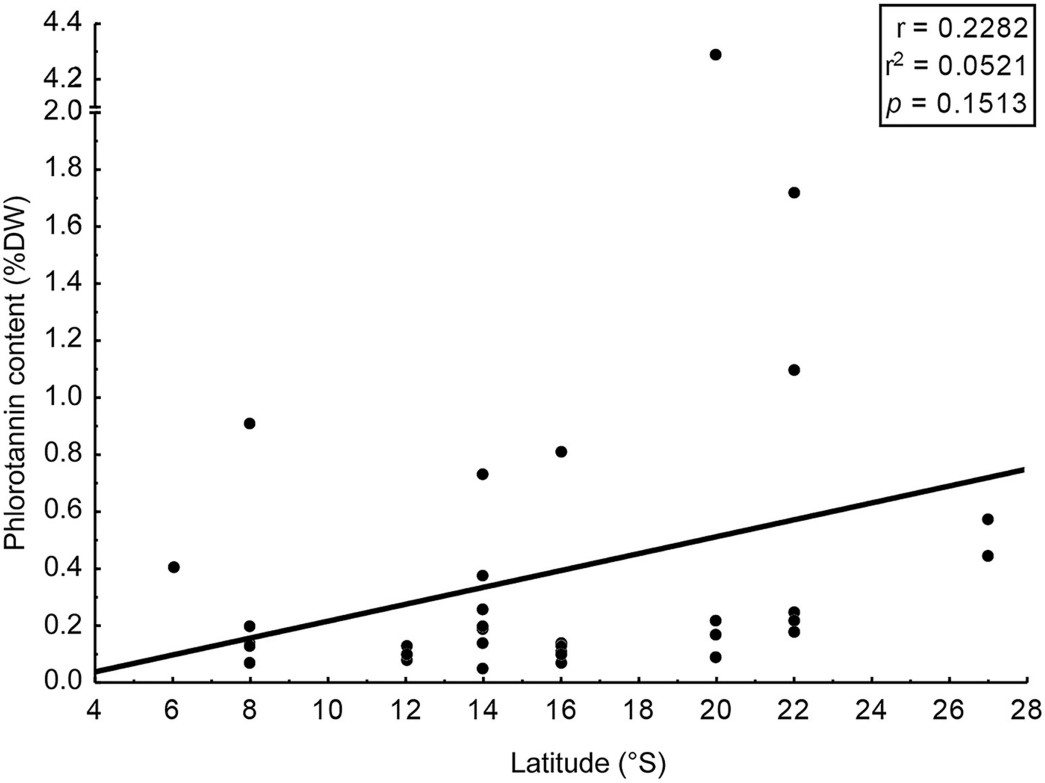

**Figure 3 Correlation: total content of phlorotannins x latitude.** Correlation between total content of phlorotannins found in the brown seaweeds and latitude (sampling site of the seaweeds).

## Variation in phlorotannin contents within populations and across a latitudinal gradient

Intra-populational analyses were carried out for 25 populations (Table 2) of 14 seaweed species (Table 1). For all analyzed populations, we identified a significant difference in the amount of phlorotannins among the individuals that comprised them ($t$-test, $p < 0.01$), with CV ranging from 5.2% to 65.3% (Table 2). CV was higher in populations collected from higher latitudes, but the correlation though significant ($p < 0.005$) was relatively weak ($r = 0.55$) (Fig. 2).

We assessed phlorotannin contents in brown seaweeds sampled along a broad latitudinal range, from 2° to 22° of southern latitude, representing from Recife to Rio de Janeiro, respectively (Table 2). The highest phlorotannin contents were found in brown seaweeds collected at higher latitudes, but the correlation between amounts and latitude was weak and non-significant (Fig. 3, $r = 0.23$; $p = 0.15$).

## DISCUSSION

The phlorotannin contents found in the brown seaweeds we investigated were typically very low (<2.0% DW), with only one exception, *Spatoglossum schroederi* for which we recorded 4.30% DW. These results reinforce a pattern that seems to be typical of tropical areas, including the Brazilian coast, in which low values of phlorotannins have

been reported for several brown seaweeds belonging to different orders, ranging from 0.2% to 2.17% DW (*Pereira & Yoneshigue-Valentin, 1999*; *Pereira et al., 1990*; *Fleury et al., 1994*). Low contents of these chemicals, varying from 0.19% to 1.62% DW, were also found in some brown seaweeds from Guam and neighboring areas of the tropical Pacific (*Steinberg & Paul, 1990*; *Van Alstyne & Paul, 1990*). Moreover, low levels of phlorotannins (ranging from 0.2% to 1.77% DW) have been found in *Sargassum* spp. and *Turbinaria* spp. at two tropical sites, Tahiti and the Great Barrier Reef, Australia, respectively (*Steinberg, 1986*).

Brown seaweed phlorotannins have been reported as defensive chemicals against herbivores in some studies (*Jormalainen & Ramsay, 2009*), but only when they occur at concentrations higher than 2.0% DW, that is, levels commonly found in species from temperate regions (*Ragan & Glombitza, 1986*). However, the evidence for this defensive property of phlorotannins remains disputed, with reports supporting (*Van Alstyne & Paul, 1990*) and refuting (*Steinberg & Paul, 1990*) this role. The low levels of phlorotannins in tropical seaweeds may be due to these chemicals having limited impact on tropical fish herbivory, given that fishes from the Great Barrier Reef do not consume more phenolic-poor tropical species than phenolic-rich species (*Steinberg & Paul, 1990*). However, contradicting this latter finding, phlorotannin-rich seaweeds were not consumed by fishes in Guam (tropical Pacific region), though extracts from phlorotannin-poor species were also not eaten (*Van Alstyne & Paul, 1990*). Moreover, phlorotannins in amounts higher than those usually found in the Brazilian brown seaweed *Sargassum furcatum* can inhibit herbivory (*Pereira & Yoneshigue-Valentin, 1999*). However, according to our results, almost all of the seaweeds we studied probably do not employ this kind of chemical defense to prevent herbivory, since phlorotannin contents were usually lower than 2.0% DW.

The hypothesis of a latitudinal gradient of phlorotannin contents is based on the assumption that herbivory pressure increases with decreasing latitude and that production of seaweed chemical defenses is selected by the action of herbivores. Accordingly, defensive chemicals should be more common and effective in tropical seaweeds. Although chemical defenses are commonly associated with herbivore abundance and pressure, no study has conclusively demonstrated that herbivores impose selective pressures on the production of secondary metabolites (*Van Alstyne & Paul, 1990*). Moreover, phlorotannins may be present in brown seaweeds for reasons other than herbivore defense, since they have been suggested to exhibit other ecological roles, such as protecting against short-wave UV radiation (*Pavia et al., 1997*), and as anti-fouling agents (*Plouguerné et al., 2010*, *2012*).

It would be difficult to establish a clear correlation between the latitudinal variability in phlorotannin production by brown seaweeds solely with the different pressures of herbivory along the Brazilian coast, even knowing that this kind of variation exists and that the seaweeds we studied were collected from a broad latitudinal range (ca. 21°). Importantly, it remains controversial if herbivory pressure selects for chemical defense production (*Pereira & Da Gama, 2008*), even across a global tropical-temperate latitudinal gradient or along the Brazilian coast (*Longo, Ferreira & Floeter, 2014*). In addition, it is known that concentrations of secondary metabolites may vary according to temperature (*Sudatti et al., 2011*), nutrient availability (*Puglisi & Paul, 1997*), light

(*Pavia et al., 1997*), salinity (*Kamiya et al., 2010*; *Sudatti et al., 2011*), and herbivory (*Weidner et al., 2004*). Thus, since the seaweeds we studied are also subjected to unknown variability in all these external conditions, it is perhaps not surprising that we did not establish a direct causal effect between phlorotannin content and latitude.

The extent of genetic control over chemical defense production remains poorly understood. For example, phlorotannin content was demonstrated to be due to genotypic variation in *Fucus vesiculosus* (*Jormalainen et al., 2003*; *Jormalainen & Honkanen, 2008*; *Koivikko et al., 2008*), as well as for terpenes in the red seaweeds *Laurencia nipponica* (*Masuda et al., 1997*; *Abe et al., 1999*) and *Delisea pulchra* (*Wright et al., 2004*). If phlorotannin production is genetically modulated, geographic distance and gene flow would likely contribute to the variation in the content of these phenols in our studied species. In general, seaweeds are considered poor dispersers because their gametes and spores only survive for a few days in the water column (*Santelices, 1990*; *Sosa & Garcia-Reina, 1993*). Limited gene flow has been reported for diverse seaweed species (*Wright, Zuccarello & Steinberg, 2000*; *Faugeron et al., 2001*, *2004*; *Zuccarello, Sandercock & West, 2002*; *Van der Strate et al., 2003*), and small-scale dispersal distances are a significant factor in the differentiation of seaweeds (*Tatarenkov et al., 2007*). Thus, if secondary metabolite production is an inherited character, geographic distance should act as a barrier to gene flow and give rise to quantitative differences in phlorotannin production.

Abiotic differences among collection sites could also support the hypothesis that different field conditions contribute to the between-site variability in phlorotannin concentrations for each of the algal species we studied. Temperature is a determining factor for the survival, geographic distribution, and reproduction of seaweeds (*Padilla-Gamiño & Carpenter, 2007*), and it is also responsible for many responses of their primary metabolism, such as photosynthesis, growth (*Nishihara, Terada & Noro, 2004*), nutrient absorption (*Tsai et al., 2005*), and secondary metabolism (*Sudatti et al., 2011*). Thus, given the reduced gene flow known for seaweeds (*Wright, Zuccarello & Steinberg, 2000*) and the different environmental conditions along the Brazilian littoral coast, populations of the same species we studied here could be highly structured, explaining in part the results we obtained. Accordingly, our field data reinforce the idea that genetic heterogeneity contributes to quantitative variation of secondary metabolism and that our sampled populations may represent ecotypes.

The intra-populational variability in the amounts of defensive chemicals we report here corroborates the findings of the few previous studies that investigated this topic in the red seaweeds *Portieria hornemannii* (*Matlock, Ginsburg & Paul, 1999*), *Delisea pulchra* (*Wright, Zuccarello & Steinberg, 2000*) and *Laurencia dendroidea* (*Sudatti, Rodrigues & Pereira, 2006*). However, those studies did not assess as broad a latitudinal context as we did. Our study also reinforces the importance of analysis at the intra-population level (i.e., variation among specimens), since most studies of seaweed chemical ecology overlook this element of chemical variation by examining pooled extracts and/or substances obtained from groups of individuals. Developmental (*Bowers & Stamp, 1993*), environmental (*Agrell, McDonald & Lindroth, 2000*), and genetic (*Berenbaum & Zangerl, 1992*) traits all represent sources of variation that can explain the diversity of plant chemical phenotypes. Moreover, in

seaweeds, life-history phases (see *Verges, Paul & Steinberg, 2008*), ontogenetics (*Paul & Van Alstyne, 1988*), and chemical races (*Abe et al., 1999*) may also be included as sources of secondary metabolite variability. In our analysis, the specimens belonged to the sporophytic life-history phase and were approximately of the same size. However, we cannot rule out the possibility that chemical races exist among the individuals of each population we studied.

## CONCLUSION

Overall, our results show that latitude does not explain the variability in total amounts of phlorotannins found in each population of the brown seaweeds we studied along the Brazilian coast, but the significant intra-specific differences in production of these chemicals we report may be important to understanding the ecological drivers of this defensive chemistry in seaweeds. Based on characteristics of the Brazilian coast (*Floeter & Soares-Gomes, 1999*), the higher phlorotannin levels we recorded in populations from higher latitudes may represent a greater capacity for these seaweeds to respond to seasonal stimuli. Since environments in low latitudes exhibit little seasonal variation, the need for seaweeds in these zones to vary production of these chemicals may be lessened. Thus, brown seaweeds at higher latitudes are more likely to modulate chemical defense production in response to stimuli than those in tropical regions where the environmental conditions are more constant. However, we assert that further studies of intra-populational variability in chemical defense are warranted in the context of marine chemical ecology.

### Funding

This work was supported by Conselho Nacional de Desenvolvimento Científico e Tecnológico (CNPq) and Fundação de Amparo à Pesquisa do Estado do Rio de Janeiro (FAPERJ). There was no additional external funding received for this study. The funders had no role in study design, data collection and analysis, decision to publish, or preparation of the manuscript.

### Grant Disclosures

The following grant information was disclosed by the authors:
Conselho Nacional de Desenvolvimento Científico e Tecnológico: CNPq.
Fundação de Amparo à Pesquisa do Estado do Rio de Janeiro: FAPERJ.

### Competing Interests

The authors declare that they have no competing interests.

### Author Contributions

- Glaucia Ank conceived and designed the experiments, performed the experiments, analyzed the data, prepared figures and/or tables, authored or reviewed drafts of the paper.

Peer J

- Bernardo Antônio Perez da Gama conceived and designed the experiments, analyzed the data, authored or reviewed drafts of the paper.
- Renato Crespo Pereira conceived and designed the experiments, analyzed the data, contributed reagents/materials/analysis tools, authored or reviewed drafts of the paper, approved the final draft.

## Field Study Permissions

The following information was supplied relating to field study approvals (i.e., approving body and any reference numbers):

Field experiments were approved by the Instituto Chico Mendes de Conservação da Biodiversidade (Authorization Number 27001-2).

## Data Availability

The raw data is available as a Supplemental File.

## Supplemental Information

Supplemental information for this article can be found online at http://dx.doi.org/10.7717/peerj.7379#supplemental-information.

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
