# Peer review of "Latitudinal variation in phlorotannin contents from Southwestern Atlantic brown seaweeds"

_PeerJ, doi:10.7717/peerj.7379_

## Round 0.1 · original submission · Major Revisions

Dear Dr. Renato C Pereira,

Thank you for submitting your manuscript to PeerJ. I apologise for the delay in getting back to you, but it has been difficult to find suitable reviewers. Finally, your manuscript entitled “Phlorotannins in brown seaweeds vary more between than within populations according to latitude” has now been evaluated by two peer reviewers, and the reviewer comments are appended below.

Your manuscript was given two thoughtful reviews. Reviewer #1 thinks the topic is interesting but the manuscript is currently rather descriptive of such processes rather than being written as a hypothesis driven study as there are several problems related to the design (e.g. lacking of data on environmental conditions and some statistical analyses) and the interpretation of the results about potential causes of the variations of the phlorotannins concentrations. Reviewer #2 considers the data in this manuscript are valuable although he/she has raised points that need to be addressed such as a clearer definition of the hypothesis and changes in the material and methods. Both reviewers consider the manuscript needs a thorough revision for correcting typos.

Based on the referees' recommendations, I arrive at this decision: The manuscript does merit publication in PeerJ but it is not acceptable in its current form and needs a major revision based on the reviews and my general comments below. I therefore invite you to resubmit a revised version, taking into account all the points raised with a special focus on problems related to the lacking of data and statistical analyses. Please carefully consider the comments of the reviewers and provide a point-by-point response which clearly defines the changes made. Please also have the language in your manuscript checked by a native English-speaking colleague.

Thank you for your patience with the evaluation process and for choosing PeerJ.

I look forward to receiving your revised manuscript.

Yours sincerely,


Blanca Figuerola
* * *
Academic editor
PeerJ
* * *
Reviewer 1 ·

Basic reporting

I find the theme interesting and valid to study, however, the paper is currently rather descriptive of such processes rather than being written as a hypothesis driven study, also, there are several problems related to the design and the interpretation of the results. Furthermore, the authors could link the different aspects phlorotannins production and the enviromental conditions. The paper also needs a thorough revision for improving the quality of English and correcting typos.

Experimental design

It is necessary to differentiate between intertidal and subtidal algae, in order to understand the processes that affect the different algae in the of phlorotannins production. Furthermore, seasonal measurements and environmental conditions, such as water temperature, salinity and / or incident light, must be carried out in order to make consistent correlations. Authors must standardize the sample number

Validity of the findings

This study is only a photograph of the phlorotannins concentrations in certain brown algae, and although the manuscript speculates on the potential causes of the variations, there is no experimental support that allows such affirmations

Additional comments

I find the theme interesting and valid to study, however, the paper is currently rather descriptive of such processes rather than being written as a hypothesis driven study, also, there are several problems related to the design and the interpretation of the results. This study is only a photograph of the phlorotannins concentrations in certain brown algae, and although the manuscript speculates on the potential causes of the variations, there is no experimental support that allows such affirmations. The authors could link the different aspects phlorotannins production and the enviromental conditions. The paper also needs a thorough revision for correcting typos.

Reviewer 2 ·

Basic reporting

In general, the writing manuscript is clear, and the literature references are relevant and sufficient.

Throughout the text there are many instances of unseparated words, which should be separated. Please correct! Some examples include lines 21,32, 36, 37, 39, 40, 42, 51, 66, 70, 72, 77, 78, 79, 82, 85-9, 93, 96, 99, 103, 105-113, 116, 123-4, 127, 141, 144, 148, 150-1, 155, 157, 159, 161-2, 164-167, 170, 173, 179, 180-3, 188, 190, 198, 204-5, 209, 217, 221-3, 229, 238, 242-5, 250, 253, References list.

Abstract: There are few changes in the English to make. Please, also, check all manuscript.
Line 18 - Instead of “littoral”, change to “coast”
Line 20 – Instead of “dry weight of the species” change to “dry weight for the species”
Line 20 - Instead of “reaffirming a typical amount of ….” change to “confirming reports of the typical amounts of…”
Line 21 - Instead of “chemicalsin” change to “chemicals in”
Line 23 – Instead of “..analyzed were verified significant differences on the amount of phlorotannins in their different individuals...” change to “analyzed there were verified significant differences on the amount of phlorotannins in different individuals..””
Line 24 - Instead of “coefficient of variation (CV)…” change to “coefficients of variation (CV)…”
Line 26 - Instead of “these aspects have been found…” change to “these variables were found…”
Lines 28/29 - Instead of “probably could easily respond to..” change to “probably as a response to..”

Introduction: The introduction is clear and relevant. It provides the justification for your study. I suggest you to include a clearer definition of the hypothesis.
Line 109 - Instead of “D. mertensii(Martius) Kützing change to “D. mertensii (C. Martius) Kützing”.
Line 111 – Instead of “S. stenophyllumMartius, S. ramifoliumKützing” change to “S. ramifolium Kützing, S. stenophyllum C. Martius”


The Figures and tables were included, as well as the raw data, however the words in the raw data should be translated to english.

Experimental design

Material and Methods
This is an original primary research. The methods are described with sufficient detail and information to permit replication, however, the time of year of collection of each population at each site should also be included. Also, a better explanation about the “small specimens analyzed” should be given.

Figures and Tables are also relevant. Some details should be improved, for instance:
Table 1.
1. Line 2 - Instead of “collect places”, change to “collection places”.
2. In the table column “seaweed species” should include the alphabetical order in the list (Sargassum ramifolium should come before S. stenophyllum)
3. Compare Table 1 and line 154 of the manuscript. There is a mistake? D. delicatula should be described as local Guarapari - ES and not Armação dos Búzios. Please check!
4. Compare Table 1 and line 178 of the manuscript. There is a mistake? D. polypodioides should be described as local Porto Seguro – BA, however it is marked only in Guarapari- ES Please check!
Table 2.
1. Line 3 - Instead of “collect sites” change to “collection sites”
2. Line 3 – should include the word “coordinates” after ..” including the …”

Validity of the findings

Impact and novelty assessed. Conclusive results. Data is robust, statistically corrected.

The conclusions support the results.

Line 173 – concerning the site Ilhéus : Compare the line 173 - Ilhéus – Back Door, and in Table 1 Ilhéus – BA.. Which is correct?

Additional comments

There is no question that the theme as such has great relevance, includes original data, and must be published. However, there are some changes that must be made. Please see the above.

---

## Round 0.2 · Minor Revisions

Dr. Renato C Pereira,

You have done a good job addressing all the comments and changes made by two reviewers.

However, a final check by one of the Section Editors has detected that the English language should be given an additional edit before Acceptance. They commented that the language in the first half of the paper is good, but in particular the Discussion still has a lot of editing needed. They are recommending that you use an English-speaking colleague or editorial service to do a revision of the language before resubmitting.

Sincerely,

Blanca Figuerola
* * *
Academic editor
PeerJ
* * *
[]

---

## Round 0.3 · Minor Revisions

Dear Renato,

Thank you very much for submitting a new version checked by an editorial service.

Before accepting the manuscript for publication I would like you address some questions made by 'John' (who is identified in the tracked changes as the person who edited your manuscript):

Lines 385-392: Please, add the year of description of the species.

Line 400: Please, clarify the meaning of 1 N FC.

Thank you very much.

I am looking forward to receiving the new version.

Sincerely,

Blanca Figuerola
* * *
Academic editor
PeerJ

---

## Round 0.4 · accepted · Accept

Dear Renato,

Thank you for clarifying these points. I am pleased to inform you that your paper has been accepted for publication without further changes.

Thank you for submitting your work to PeerJ. We hope you consider us again for future submissions.

Best regards,

Blanca Figuerola
Academic Editor, PeerJ